# Topological analog signal processing

Farzad Zangeneh-Nejad [1] & Romain Fleury [1]

Analog signal processors have attracted a tremendous amount of attention recently, as they potentially offer much faster operation and lower power consumption than their digital versions. Yet, they are not preferable for large scale applications due to the considerable observational errors caused by their excessive sensitivity to environmental and structural variations. Here, we demonstrate both theoretically and experimentally the unique relevance of topological insulators for alleviating the unreliability of analog signal processors. In particular, we achieve an important signal processing task, namely resolution of linear differential equations, in an analog system that is protected by topology against large levels of disorder and geometrical perturbations. We believe that our strategy opens up large perspectives for a new generation of robust all-optical analog signal processors, which can now not only perform ultrafast, high-throughput, and power efficient signal processing tasks, but also compete with their digital counterparts in terms of reliability and flexibility.

[1] Laboratory of Wave Engineering, Swiss Federal Institute of Technology in Lausanne (EPFL), 1015 Lausanne, Switzerland. Correspondence and requests for materials should be addressed to R.F. (email: romain.fleury@epfl.ch)

For a few decades, digital signal processors (DSPs) have widely replaced analog electronic and mechanical computers for carrying out computational tasks. Such processors not only offer high-speed operation, but also ensure reliability and flexibility of the processing[1], a property which has established DSPs as perfect candidates for realizing large-scale computational systems.

While providing high-speed and reliable operation, DSPs also suffer from several fundamental drawbacks such as high-power consumption, costly analog to digital converters, and drastic performance degradation at high frequencies[2]. Considering these limitations, it is neither reasonable nor affordable to use DSPs for performing specific, simple computational tasks such as differentiation or integration, equation solving, matrix inversion, edge detection and image processing. Therefore, the old idea of all-analog computing and signal processing has been recently revived, driven by the development of cost-efficient nanofabrication techniques and promising related advances in ultrafast optics. In their pioneer work, Silva et al.[3] theoretically demonstrated the possibility of carrying out simple computational tasks such as convolution, differentiation and integration, making use of optical waves as they propagate through engineered metamaterial layers. By going beyond the aforementioned restrictions of DSPs, such a wave-based computational scheme then inspired numerous exciting applications including analog computing[4–15], signal processing, equation solving[16, 17], optical image processing[18–21], optical memories[22], and photonic neural networks[23]. Not only are such types of analog signal processors (ASPs) real time and ultrafast due to their wave-based nature, but also they offer low power consumption and high-throughput operation as they are free of analog/digital conversion steps. In addition, they allow the unique possibility of carrying out different computational tasks in parallel, thereby significantly reducing the total processing time[24–30].

Despite their advantageous properties, ASPs still suffer from one important limitation compared to DSPs, which severely hinders their applicability for large-scale applications: while repeating the same operation always gives rise to the same result when using DSPs (which is enabled by available error-finding algorithms and protocols in digital systems), analog signal processing is often accompanied with considerable observational error caused by the extreme sensitivity of ASPs to changes in environmental and structural parameters[31].

Here, we demonstrate the possibility of drastically enhancing the reliability of ASPs by leveraging the unique immunity of topological insulators[32–47] against imperfections, a much-sought feature which has established itself as a new paradigm for realizing a large variety of reliable devices such as lasers[48], modulators[49], and lenses[50]. More specifically, we demonstrate a topological wave-based analog system that can solve linear differential equations of arbitrary order in time-domain at the speed of the wave and prove its strong immunity against geometrical flaws and environmental changes. Our findings provide exciting perspectives for a new generation of ultrafast, high-throughput, and highly reliable analog computing systems and signal processors.

## Results

**Topological analog equation solver.** To illustrate the core idea of our proposal, let us consider an important signal processing task, namely the resolution of a first-order linear differential equation with a non-zero arbitrary forcing $g(t)$ (below, we will generalize the concept to differential equations of higher order). The goal is to design a two-port wave system that, regardless of the specific form of $g(t)$ sent at the input, outputs a signal $f(t)$ that would be the solution of a differential equation $f'(t) + \alpha f(t) = \beta g(t)$, where $\alpha$ and $\beta$ are real coefficients. To this aim, one could consider a conventional resonator created by, for example, a one-dimensional photonic band gap material with a defect in the middle (Fig. 1a). According to coupled-mode theory, the transmission coefficient through such resonator near its resonance frequency $f_0$ is given by[51]

$$H(f) = \frac{A}{j(f - f_0) + f_0/2Q}, \quad (1)$$

where we have used the time harmonic convention $\exp(j2\pi f t)$, $A$ is an arbitrary constant, and $Q$ is the quality factor of the resonance. Now, if we consider as input signal the source term $g(t)$ modulated at the carrier frequency $f_0$, i.e. $\tilde{g}(t) = g(t)\cos(2\pi f_0 t)$, its relationship with the output $\tilde{f}(t) = f(t)\cos(2\pi f_0 t)$ can be obtained via inverse Fourier transform of the transfer function (TF) $H(f)$ of Eq. 1, leading to the desired first-order differential equation $f'(t) + \alpha f(t) = \beta g(t)$, with $\alpha = \pi f_0/Q$ and $\beta = 2\pi A$. This analysis illustrates the possibility to realize an analog equation

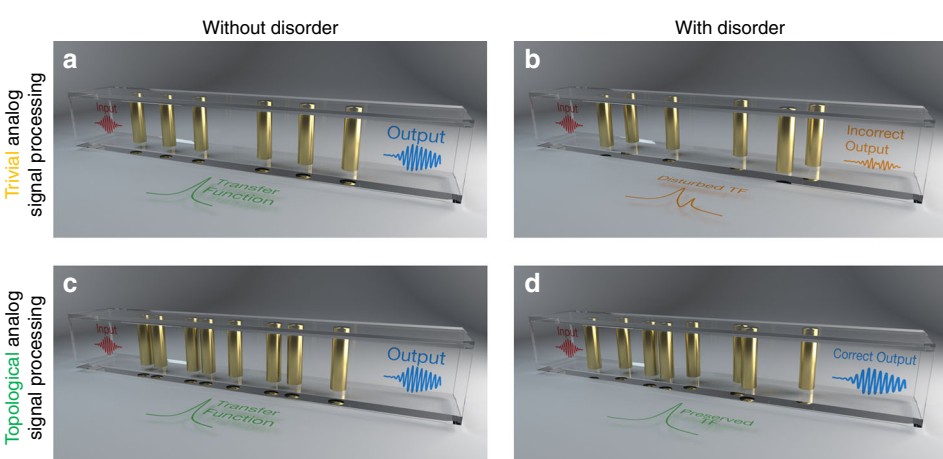

**Fig. 1** Robust topological analog signal processing. We consider the possibility to process time-domain wave signals by engineering their transfer function as they propagate through an engineered solver. **a** A first-order differential equation solver is constructed from resonant tunneling through a crystal defect. The output signal is the solution of the differential equation associated with the transfer function of the system. **b** In the presence of geometrical defects, like slight position shifts, the signal processing functionality achieved with the trivial equation solver of panel (**a**) is completely destroyed. **c** To make the signal processing robust, we propose instead to build the target transfer function of the system from resonant tunneling through a topological edge mode. **d** Markedly different from the trivial equation solver of panels (**a**) and (**b**), the output of the topological solver is left totally unaffected by the disorder

solver by engineering a resonator and measuring the envelope of its response to the known input signal $g(t)$ (modulated at the resonance frequency). From this example, the main advantages of analog computation over digital one are evident: there is no need for converting the input signal $g(t)$ to a digital stream (and vice versa), and the computation is being carried out in real time with no limitation at large values of $f_0$. However, this simplicity comes with a severe drawback: when adding geometrical imperfections to the system (Fig. 1b), the transfer function can be significantly disturbed by the creation of disorder-induced modes, shifting its spectrum and introducing new resonating peaks. This leads to an output signal which has nothing to do with the correct solution.

What we propose is instead to form the same transfer function, but out of resonant transmission through a topological edge mode, whose existence is guaranteed by the nontrivial topologies of the surrounding bulk insulators. Figure 1c depicts such a solution based on two insulating lattices with supposedly different topologies, inspired by the Su−Schrieffer−Heeger (SSH) scheme[32]. Like the previous case, when the input signal $g(t)$ (modulated at $f_0$) is applied to such system, the output signal envelope $f(t)$ is equal to the solution of the desired differential equation. However, the topological equation solver can be immune to disorder, since the presence of a single mid-gap interface mode can be guaranteed by bulk-edge correspondence[52] (Fig. 1d).

To test this idea on a realistic system, we designed a topological first-order linear differential equation solver for airborne audible acoustic signals (see Supplementary Note 1 for an electromagnetic equivalent). The topological ASP system is based on sonic topological insulators inspired by the SSH scheme, obtained from solid cylinders placed in a pipe of square cross-section, as in Fig. 1c, d (see Methods for geometrical details and topological invariant calculations). The topological interface is designed to provide a resonant mode at $f_0 = 2254\,\mathrm{Hz}$, with $A = 1$ and $Q = 0.5f_0$, aiming at solving the differential equation $f'(t)+2\pi f(t) = 2\pi g(t)$. The transfer function $H(f)$ of the system, calculated by three-dimensional full-wave finite-element calculations in the frequency domain, is compared to the target transfer function in Fig. 2a (green and dashed curves in the middle inset), revealing their perfect agreement. Now, consider an input signal $\tilde{g}(t)$ with an arbitrarily chosen time envelope $g(t)$ to be injected into the waveguide (Fig. 2a, left). The corresponding output signal $\tilde{f}(t)$ is then calculated by convoluting $\tilde{g}(t)$ with the impulse response of the system, obtained from $H(f)$ (we have also verified our results by direct simulations in the time domain, see Supplementary Note 2). Comparing the envelope of the resulting output signal $f(t)$ (blue line) to the exact solution of the intended differential equation (dashed line) reveals that the topological ASP system is indeed solving the equation as sound propagates through the system.

Next, we add some disorder to our equation solver by randomly shifting the position of the cylinders (average position shift is 18% of lattice period in any direction) and repeat the same procedure in the bottom panel of Fig. 2a. We notice that, despite the relatively large level of disorder, the transfer function $H(f)$ has been left almost unaffected. Hence, the corresponding output signal $f(t)$ still corresponds to the solution of the desired differential equation, confirming the high robustness of the proposed equation solver. To demonstrate that this property is indeed linked to the topological nature of the system, we repeat this analysis for a topologically trivial equation solver, which is based on a resonance induced by defect-tunneling through a Bragg band gap. As confirmed in Fig. 2b, such resonating system is also capable of solving the first-order differential equation. The transfer function $H(f)$, and the output signal $\tilde{g}(t)$ is however

severely affected when imparting similar imperfections to the sample (position shifts have the same magnitude as that of topological case). This clearly affirms the superiority of topological ASPs over trivial ones. It should be pointed out that the choice of the input signal envelope $g(t)$ is arbitrary here and any other temporal form can be considered for $g(t)$ (see Supplementary Note 3).

**Symmetry protection of the topological equation solver**. Since one-dimensional topological phases are symmetry protected, these numerical results raise an important question: what is the underlying symmetry of the proposed system that protects its edge modes? In regular tight-binding SSH chains, made of evanescently coupled identical resonators with detuned hoppings $K$ and $J$ (Fig. 3a, top panel), the mid-gap edge mode occurring at the topological boundary is protected by chiral symmetry, and a transfer function based on tunneling through this edge mode is robust to disorder in the hoppings, as long as they are weak enough not to close the band gap. However, transmission is not robust to even small levels of on-site disorder, which breaks chiral symmetry. This is exemplified in Fig. 3a–c. Figure 3a shows the mid-gap spectral transmission resonance associated with a perfectly ordered sample. Figure 3b shows the transfer function immunity to disorder in the couplings. Finally, Fig. 3c shows the large sensitivity of the transmission peak to arbitrarily small disorder in the resonance frequencies, which breaks the chiral symmetry. Our multiple scattering system, albeit not based on evanescent coupling, behaves similarly. The transmission peak of the ordered sample (Fig. 3d) survives disorder shifts that do not close the band gap (Fig. 3e), but not disorder in the obstacle radii (Fig. 3f). These numerical results, obtained from full-wave finite-elements simulations, are fully consistent with our topological theory, which defines topological invariants on each bulk band using the unit cell transfer matrix $M_{\mathrm{cell}}$, that maps the Brillouin circle to a subspace of $SU(1,1)$ matrices (see Methods). Remarkably, the topological invariants can only be defined under the symmetry $M_{\mathrm{cell}}^2 = 1$, which holds for position disorder, but not for radii disorder, as we prove in Methods.

**Experimental demonstration**. Based on these findings we have built a prototype of the topological equation solver (Fig. 4a, top signal path). We first perform a frequency-domain measurement to obtain the transfer function of the system, $H(f)$, by exciting the waveguide with pseudo-random noise and recording the transmitted pressure with a microphone. The graph in the middle inset represents the magnitude of the measured transfer function (green curve) compared to what we get from the numerical simulations (gray curve). As observed, the transfer function has a peak near the resonance frequency $f_0$ of the topological edge mode, corresponding to the resonance parameters $A = 0.87$ and $Q = 0.03f_0$, or differential equation parameters $\alpha = 2.7\pi$, $\beta = 10\pi/3$. We next switch to a direct time-domain experiment and inject the same arbitrary input signal $\tilde{g}(t)$ as in Fig. 2 into the waveguide (see Supplementary Note 4 for measurements with other types of input signals). Comparing the measured transmitted pressure $\tilde{f}(t)$ (blue line) with the exact solution of the corresponding differential equation (dashed line) confirms the proper functioning of the equation solver. To probe its stability, we then randomly move the cylindrical scatterers and repeat the same procedure (Fig. 4a, bottom signal path). Remarkably, the topological ASP is still perfectly functional despite these large shifts. This exceptional property is strikingly highlighted when we compare the measured output signal from the topological equation solver with that measured at the output of its trivial counterpart in the presence of

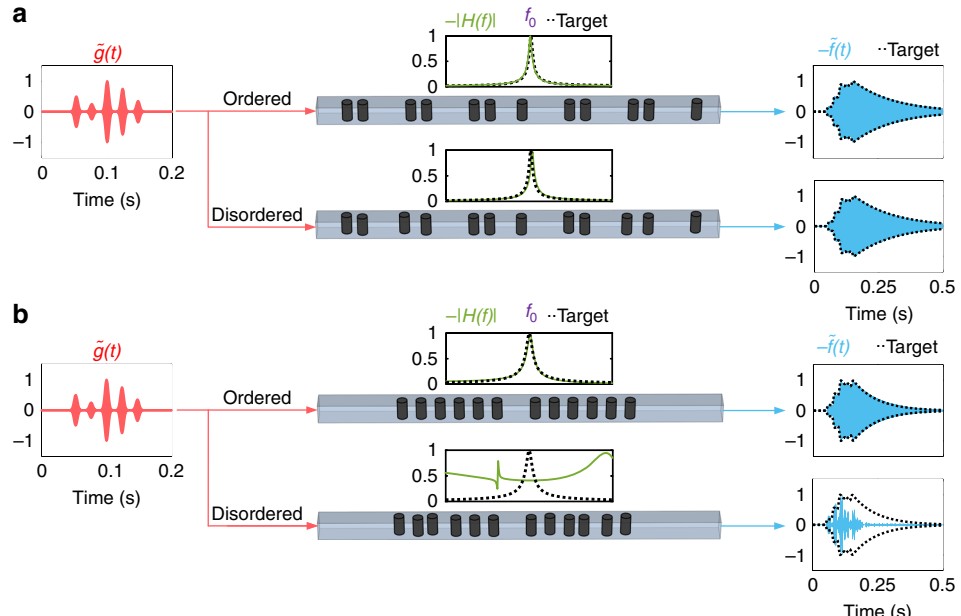

**Fig. 2** Numerical demonstration of the topological differential equation. We compare a topological (**a**) and a trivial (**b**) acoustic signal processors designed such that the envelope of their output $\tilde{f}(t)$ is the solution of the differential equation $f'(t) + \alpha f(t) = \beta g(t)$, where $g(t)$ is envelope of the input signal $\tilde{g}(t)$ modulated with carrier frequency $f_0$. **a** An arbitrarily chosen signal envelope $g(t)$ is applied to the input of the topological equation solver. The transfer function of the system $H(f)$ (green line), which reproduces exactly the mathematical target defined by the equation (dashed line), is not affected by the presence of disorder (bottom signal path). As a result, the envelope of the output signal $f(t)$ matches exactly the solution even in the presence of disorder. **b** Conversely, in a topologically trivial processor, the presence of disorder-induced localized states creates spurious peaks and shifts the transfer function of the system, which makes it deviate from the targeted transfer function (dashed line). The parameters of the linear differential equation are chosen to be $\alpha = \beta = 2\pi$, and the position disorder strength is 18% of the lattice period in both cases

disorder of similar strength (Fig. 4b). Very different from the topological processor, the signal coming out of the trivial processor is completely distorted, which clearly validates the superior robustness of topological ASP systems. Note that a large range of values for the parameter $\alpha$ and $\beta$ can be targeted using various means, like changing the lattice periodicity, number of unit cells, and adding losses (see Supplementary Note 5).

To further extend the reach of the approach, we investigate the possibility of solving linear differential equations of higher order. Suppose, for instance, that we want to solve the second-order differential equation $f'(t) + 6\pi f'(t) + 8\pi^2 f(t) = 4\pi^2 g(t)$, which corresponds to the transfer function $H(f) = 1/(2 + 3j(f - f_0) - (f - f_0)^2)$. Using partial fraction decomposition, one can then write $H(f) = H_1(f) - H_2(f)$, with $H_1(f) = 1/(1 + j(f - f_0))$ and $H_2(f) = 1/(2 + j(f - f_0))$. It follows that in order to solve the desired second-order differential equation, we can realize two (first-order) equation solvers with the transfer functions $H_1(f)$ and $H_2(f)$ and subtract their output signals (see Fig. 5a). This is accomplished in an analog way in Fig. 5b, where $H_1(f)$ and $H_2(f)$ are realized using two different topological first-order systems with tailored dissipation losses. The analog subtraction operation is realized with an acoustic rat-race coupler. Full-wave simulations involving the full geometry with the two-pipes and the rat-race coupler confirm that $H(f) = H_1(f) - H_2(f)$ is properly implemented. Hence, when an input signal, $\tilde{g}(t)$, say for example with a Gaussian envelope, is applied to the system, the envelope $f(t)$ of the output signal follows the exact solution of the target differential equation. This is confirmed by direct finite difference time domain (FDTD) simulations.

The experimental demonstration of topological second-order differential equation solving is provided in Fig. 5c. We designed two first-order differential equation solvers connected to each other via our 3D-printed acoustic rat-race coupler. The two first-

order ODE solvers are tuned to solve the second-order ODE by adjusting the level of transmission losses using sound absorbing melamine foam. We then simultaneously excited both waveguides with the input signal $\tilde{g}(t)$, and measured the output $\tilde{f}(t)$. As seen in the figure, excellent agreement exists between the measured output signal envelope $f(t)$ (solid blue line) and the expected exact solution of the corresponding second-order differential equation (dashed line). This technique can easily be extended to the resolution of differential equations of arbitrary order (see Supplementary Note 6). It is worthy to mention further that, as an alternative route to what we proposed here, one can realize higher order and more complex transfer functions by cascading two or more SSH chains, allowing their topological edge modes to couple to each other, as we demonstrate in Supplementary Note 7.

## Discussion
The robustness of the proposed topological ASPs constitutes a key step towards a new generation of all-acoustic or all-optical ASPs, which not only can be much faster and simpler than DSPs, but also can compete with them in terms of reliability and flexibility. Our proposed strategy also opens new perspectives for further explorations that merge the field of topological insulators with linear system theory. For instance, while we focused here on time-domain signal processing (solving differential equations in real time), performing topological operation in other domains such as space or frequency (i.e. topological filters) seems equally promising and straightforward from our findings (an example is given in Supplementary Note 8). Another interesting direction would be to translate the concept to optics, where general design tools such as optical circuits and metatronics[53] can enable an easy implementation of the topological concept proposed here, and

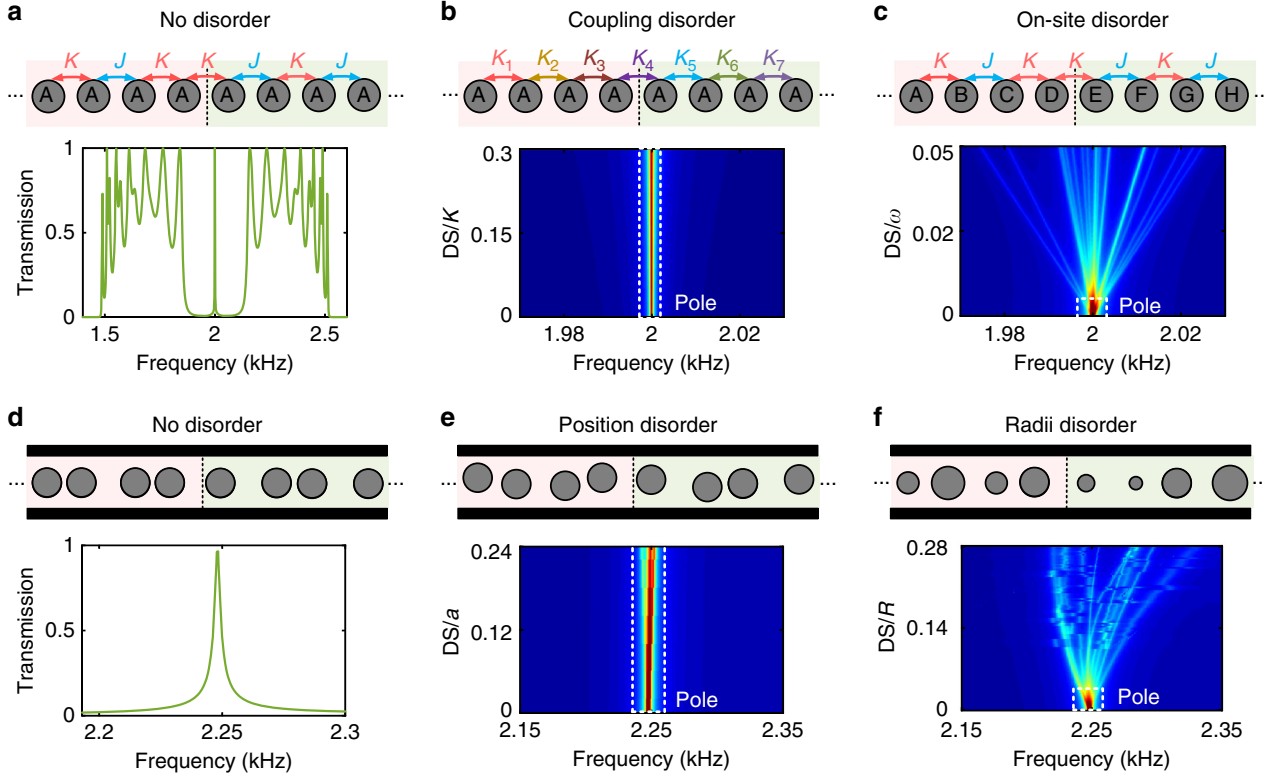

**Fig. 3** Effect of various defect types on the topological equation solver. **a** A topological interface made from tight-binding SSH chains (top), consisting of resonators with resonance frequency $\omega_0$ coupled to each other via detuned hopping amplitudes $K$ and $J > K$, supports an edge mode protected by chirality. The transmission spectrum of the chain (bottom) shows a mid-gap resonance, which corresponds to the topological edge mode. **b** Some disorder is added to the hopping amplitudes of the system (top), which preserve chiral symmetry. The bottom panel demonstrates the robustness of the transmission peak (averaged over 20 realizations of disorder) as the disorder strength (DS) is increased. **c** Same as panel (**b**) except that the disorder is applied to the on-site potentials of the chain, hereby breaking chiral symmetry. The transmission peak is sensitive to arbitrarily weak disorder. **d–f** Same as (**a–c**) but for the proposed acoustic equation solver. The resonance line-shape of the edge mode is robust to the position movement (normalized to the lattice constant) of the rods inside the waveguide (panel **e**), which does not break the symmetry $M_{\mathrm{cell}}^2 = 1$ (see Methods). In contrast, detuning the radii of the obstacles breaks this property, and causes degradation in the performance of the equation solver (panel **f**)

facilitate the design and optimization of topological ASPs, so that they perform more complex signal processing tasks, or so that they become more robust to certain types of defects instead of others (immunity engineering). Alternatively, one may think of generalizing the concept to two-dimensional topological systems protected by time-reversal symmetry, which offer immunity against a broader range of defects. Finally, we envision that the coupling between topological ASPs systems and nonlinearities may lead to exciting venues for large-scale neural network systems with robust processing capabilities.

## Methods

**Bloch eigenproblem.** The bulk crystal is one-dimensional with lattice constant $a$ and two obstacles per unit cell. We model it and define its topology using the transfer matrix $M_{\mathrm{cell}}$ of a unit cell. We start by defining the two scattering matrices $S_1$ and $S_2$, as the far-field scattering matrices of each obstacle when being alone in the monomode waveguide. These matrices relate the outgoing complex signals on the left ($L$) and right ($R$) sides of the scatterers $b_L$ and $b_R$ to the incident ones, $a_L$ and $a_R$:

$$\begin{pmatrix} b_{L,i} \\ b_{R,i} \end{pmatrix} = S_i \begin{pmatrix} a_{L,i} \\ a_{R,i} \end{pmatrix}. \tag{2}$$

Note that for now we do not make the assumption that the two matrices are equal: for instance, the cylinders could have different cross-sections, or be shifted with respect to each other, etc. These matrices also usually depend on the angular frequency $\omega$. Assuming conservation of energy during the scattering process, they must be unitary. We can therefore parametrize them very generally as

$$S_1 = \begin{pmatrix} e^{i\varphi_1}\cos\theta_1 & e^{i\alpha_1}\sin\theta_1 \\ -e^{-i\alpha_1}\sin\theta_1 e^{i\Phi_1} & e^{-i\varphi_1}\cos\theta_1 e^{i\Phi_1} \end{pmatrix}, \tag{3}$$

$$S_2 = \begin{pmatrix} e^{i\varphi_2}\cos\theta_2 & e^{i\alpha_2}\sin\theta_2 \\ -e^{-i\alpha_2}\sin\theta_2 e^{i\Phi_2} & e^{-i\varphi_2}\cos\theta_2 e^{i\Phi_2} \end{pmatrix}, \tag{4}$$

where the frequency-dependent angles $\theta_{1,2}$, $\alpha_{1,2}$, $\phi_{1,2}$, and $\Phi_{1,2}$ are unique once we fix the reference plane, here at the central position of the scatterers. Assuming reciprocity ($S_{21} = S_{12}$), we must have $2\alpha_{1,2} - \Phi_{1,2} = \pi$, which restricts us to three parameters per scattering matrix, allowing to write:

$$S_1 = \begin{pmatrix} e^{i\varphi_1}\cos\theta_1 & e^{i\alpha_1}\sin\theta_1 \\ e^{i\alpha_1}\sin\theta_1 & -e^{-i\varphi_1}\cos\theta_1 e^{2i\alpha_1} \end{pmatrix}, \tag{5}$$

$$S_2 = \begin{pmatrix} e^{i\varphi_2}\cos\theta_2 & e^{i\alpha_2}\sin\theta_2 \\ e^{i\alpha_2}\sin\theta_2 & -e^{-i\varphi_2}\cos\theta_2 e^{2i\alpha_2} \end{pmatrix}. \tag{6}$$

One can then derive the associated transfer matrices $M_1$ and $M_2$, defined as

$$\begin{pmatrix} b_{R,i} \\ a_{R,i} \end{pmatrix} = M_i \begin{pmatrix} a_{L,i} \\ b_{L,i} \end{pmatrix} \tag{7}$$

and obtains

$$M_1 = \begin{pmatrix} \dfrac{e^{i\alpha_1}}{\sin\theta_1} & -\dfrac{e^{-i\varphi_1}e^{i\alpha_1}\cos\theta_1}{\sin\theta_1} \\ -\dfrac{e^{i\varphi_1}e^{-i\alpha_1}\cos\theta_1}{\sin\theta_1} & \dfrac{e^{-i\alpha_1}}{\sin\theta_1} \end{pmatrix}, \tag{8}$$

$$M_2 = \begin{pmatrix} \dfrac{e^{i\alpha_2}}{\sin\theta_2} & -\dfrac{e^{-i\varphi_2}e^{i\alpha_2}\cos\theta_2}{\sin\theta_2} \\ -\dfrac{e^{i\varphi_1}e^{-i\alpha_2}\cos\theta_2}{\sin\theta_2} & \dfrac{e^{-i\alpha_2}}{\sin\theta_2} \end{pmatrix}. \tag{9}$$

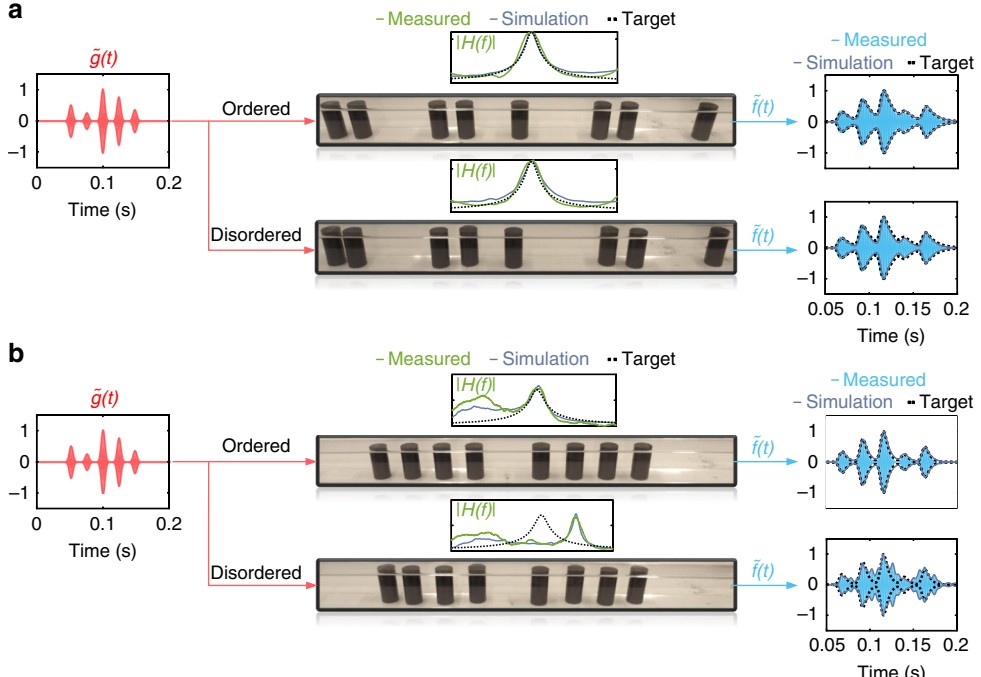

**Fig. 4** Experimental demonstration of the topological equation solver. The acoustic waveguide is a square transparent tube and the scatterers are made from black Nylon rods. As in our numerical investigations of Fig. 2, we compare the trivial signal processors in terms of robustness to position defects. A frequency domain measurement allows us to extract the transfer function $H(f)$ (green line), which we compare to the ideal target (dashed line) and simulation (gray). An additional measurement then performed in time domain by directly sending an input signal with envelope $g(t)$ into the system, and recording the coda with envelope $f(t)$ at the output. **a** The topological equation solver is indeed found to be immune to the shits in rods position. **b** Very differently, the trivial equation solver is severely affected. The parameters of the linear differential equation are chosen to be $\alpha = 2.7\pi$, $\beta = 10\pi/3$, and the position disorder has the same strength in both cases

If the two scatterers are separated by a distance $d$ in a unit cell of lattice constant $a$, the total transfer matrix of the unit cell $M_{\text{cell}}$ is the product:

$$M_{\text{cell}} = M_{\frac{a-d}{2}}M_2M_dM_1M_{\frac{a-d}{2}} \tag{10}$$

with

$$M_L = \begin{pmatrix} e^{\frac{i\omega L}{c}} & 0 \\ 0 & e^{-\frac{i\omega L}{c}} \end{pmatrix}, \tag{11}$$

where $L = d, \frac{a-d}{2}$, and $c$ is the phase velocity. One obtains, after taking the matrix product,

$$M_{\text{cell}}(\omega) = \begin{pmatrix} M_{11}(\omega) & M_{21}^*(\omega) \\ M_{21}(\omega) & M_{11}^*(\omega) \end{pmatrix} \tag{12}$$

with

$$M_{11}(\omega) = e^{\frac{i\omega a}{c}}e^{i(a_1+a_2)}\csc\theta_1\csc\theta_2 + e^{\frac{i\omega(a-2d)}{c}}e^{i(\varphi_1-\varphi_2)}e^{-i(a_1-a_2)}\cot\theta_1\cot\theta_2, \tag{13}$$

$$M_{21}(\omega) = -e^{\frac{i\omega d}{c}}e^{i\varphi_2}e^{i(a_1-a_2)}\csc\theta_1\cot\theta_2 - e^{-\frac{i\omega d}{c}}e^{i\varphi_1}e^{-i(a_1+a_2)}\cot\theta_1\csc\theta_2. \tag{14}$$

We use the notation $z^*$ to denote the complex conjugate of $z$. Noting $|\psi\rangle = [a, b]^T$, with $a$ and $b$ being the forward and backward complex field amplitudes at the entrance of the unit cell, the application of Bloch theorem yields the following eigenvalue problem,

$$M_{\text{cell}}(\omega)|\psi\rangle = e^{ik_{\text{B}}a}|\psi\rangle \tag{15}$$

which we call the Bloch eigenproblem of the crystal. Note the nontrivial dependence of $M_{\text{cell}}(\omega)$ on $\omega$. The most straightforward use of the above equation is the following way: for all values of $\omega$, one can diagonalize $M_{\text{cell}}(\omega)$, and get two opposite values $\pm k_{\text{B}}(\omega)$ of the Bloch wavenumber in the first Brillouin zone, and resolve the band structure. Note that $M_{\text{cell}}$ is not unitary and is non-Hermitian, meaning that in general, the values $\pm k_{\text{B}}(\omega)$ are complex, allowing in principle for an infinite number of bands and bandgaps. Note further the difference with the standard tight-binding SSH model, which leads to a Hermitian eigenvalue problem that maps the Brillouin circle into the space of SU(2) matrices, and a clear topological classification of chiral symmetric systems via the winding number. Here, consistent with time-reversal symmetry[54], $M_{\text{cell}}(\omega) \in \text{SU}(1,1)$, a group of non-Hermitian matrices[55]. SU(1,1) Hamiltonians are found, for instance, in PT-symmetric extensions of the SSH tight-binding model[56] where non-Hermiticity of

the Hamiltonian originates from the absence of energy conservation. Here, $M_{\text{cell}}$ is not a Hamiltonian, in the sense that its eigenvalues are not related to $\omega$, but to $k_{\text{B}}$, and the pseudo anti-Hermiticity of $M_{\text{cell}}$ ($\sigma_z M_{\text{cell}}^\dagger \sigma_z = -M_{\text{cell}}$) is related to time-reversal symmetry. In Supplementary Fig. 11 we represent the band structure obtained from the transfer matrix approach, and compare it with the one obtained directly from full-wave simulations of the unit cell subjected to periodic boundary conditions (FEM method). To solve the transfer matrix eigenvalue problem, the parameters $\theta_{1,2}$, $\alpha_{1,2}$, and $\Phi_{1,2}$, which depend on frequency, were extracted from FEM scattering simulations of a single obstacle in a waveguide. The distance between the two scatterers is taken to be $d = \frac{a}{2} - e_p$, with $e_p = 2.8$ cm ("trivial" case) and $a = 23$ cm. The rod diameter is 3.5 cm and the width of the waveguide is 7 cm. The agreement between the two approaches validates the accuracy of the multiple scattering model, in particular the underlying assumption of no near-field interactions between the obstacles in the crystal.

**Properties of the unit cell transfer matrix.** To define the topology of the system in the next section, we first need to establish a few key properties of the unit cell transfer matrix. We start with general properties, before moving to more specific properties on a band or at degenerate points of the band structure.

As a direct consequence of time-reversal symmetry[54], the transfer matrix of the system $M_{\text{cell}}$ belongs to the group SU(1,1) of matrices of the form

$$M_{\text{cell}} = \begin{pmatrix} \alpha & \beta^* \\ \beta & \alpha^* \end{pmatrix} \tag{16}$$

which is parametrized using the Pauli matrices as

$$M_{\text{cell}} = \alpha_{\text{R}}\sigma_0 + \beta_{\text{R}}\sigma_x + \beta_{\text{I}}\sigma_y + i\alpha_{\text{I}}\sigma_z. \tag{17}$$

Its eigenvalues, given by $\lambda_\pm = \alpha_{\text{R}} \pm i\sqrt{\alpha_{\text{I}}^2 - \beta_{\text{R}}^2 - \beta_{\text{I}}^2}$ are real when $\alpha_{\text{I}}^2 < |\beta|^2$, and complex otherwise. These eigenvalues are degenerate under the condition $\alpha_{\text{I}}^2 - \beta_{\text{R}}^2 - \beta_{\text{I}}^2 = 0$, i.e. when the parameters $\beta_{\text{R}}$, $\beta_{\text{I}}$ and $\alpha_{\text{I}}$ belong to a double cone in the $(\beta_{\text{R}}, \beta_{\text{I}}, \alpha_{\text{I}})$ space. This cone is represented in the bottom panels of Fig. 6. At the tip of the cone, one has $\beta_{\text{R}} = \beta_{\text{I}} = \alpha_{\text{I}} = 0$, meaning that $M_{\text{cell}}$ reduces to $M_{\text{cell}} = \alpha_{\text{R}}\sigma_0$.

On a band, the matrix $M_{\text{cell}}$ has a special form. Indeed, the Bloch eigenproblem implies that $\alpha_{\text{R}} \pm i\sqrt{\alpha_{\text{I}}^2 - |\beta|^2} = e^{ik_{\text{B}}a}$, from which follows that

$$\alpha_{\text{R}} = \cos(k_{\text{B}}a) \tag{18}$$

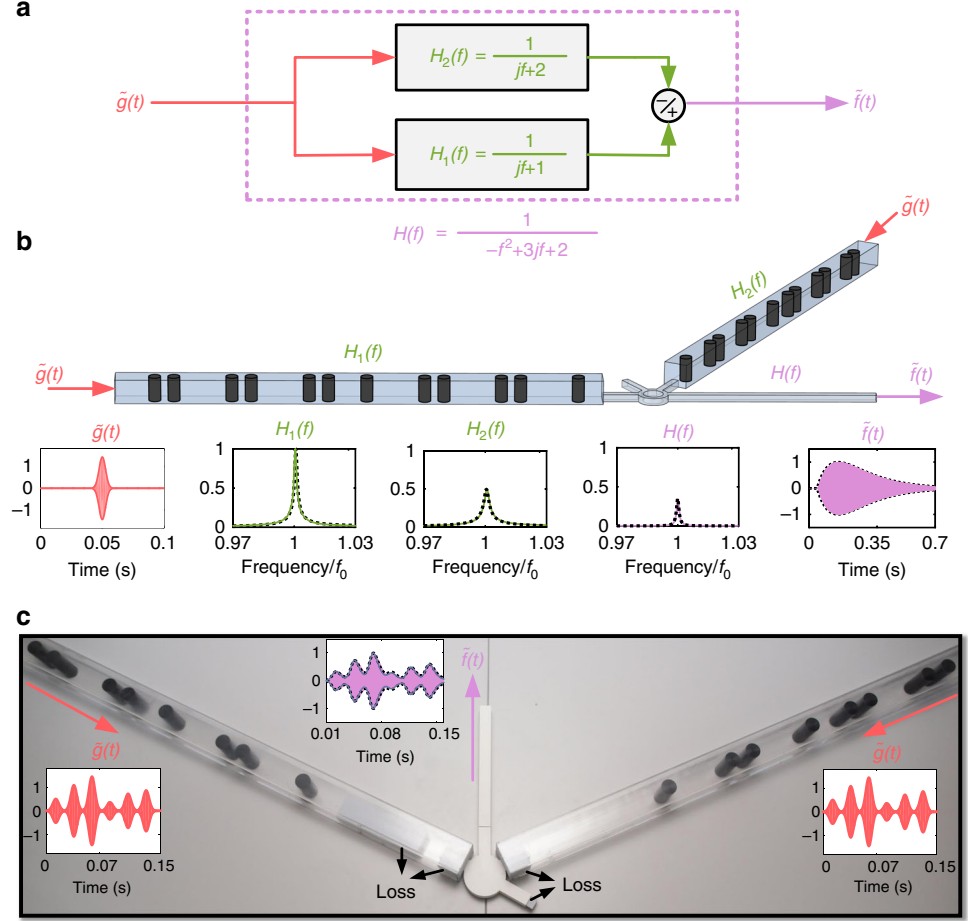

**Fig. 5** Robust resolution of a second-order differential equation. **a** The second-order transfer function associated with the resolution of the equation $f''(t) + 6\pi f'(t) + 8\pi^2 f(t) = 4\pi^2 g(t)$ can be achieved by proper subtraction of two first-order transfer functions. **b** Implementation of the scheme in panel (**a**) with two topological first-order differentiators. The signal subtraction is realized with a rat-race coupler (circular component connecting the two systems). The bottom panels represent full-wave numerical simulations of the complete 3D structure in the case of a Gaussian pulse input, demonstrating that the targeted signal processing task is indeed performed by the system. **c** Experimental realization of the second-order differential equation solver. The measured output signal envelope ($f(t)$, purple lines) is found to be in perfect agreement with both the numerical simulation (gray) and with the exact solution of the corresponding second-order differential equation (dashed line)

and

$$|\alpha|^2 = 1 + |\beta|^2 \tag{19}$$

implying $\alpha_I^2 + \alpha_R^2 = 1 + |\beta|^2$, which is equivalent to $\alpha_I^2 = \sin^2(k_B a) + |\beta|^2$, or

$$\alpha_I = \pm\sqrt{\sin^2(k_B a) + |\beta|^2}. \tag{20}$$

On a band, we therefore have

$$M_{\text{cell}} = \begin{pmatrix} \cos(k_B a) \pm i\sqrt{\sin^2(k_B a) + |\beta|^2} & \beta* \\ \beta & \cos(k_B a) \mp i\sqrt{\sin^2(k_B a) + |\beta|^2} \end{pmatrix}. \tag{21}$$

As a result, a band describes a one-to-one mapping from the Brillouin circle onto a closed path $\mathcal{C}$ in the subspace of SU(1,1) matrices $M_{\text{cell}}(k_B)$ with the above form. From the Bloch eigenvalue problem $M_{\text{cell}}(\omega)|\psi\rangle = e^{ik_B a}|\psi\rangle$, one deduces that on a band, $M_{\text{cell}}(\omega)$ has complex eigenvalues, meaning that $\alpha_I^2 > |\beta|^2$, i.e. the path $\mathcal{C}$ must be inside the cone, either in the upper region $\alpha_I > |\beta|$, or the lower one $\alpha_I < -|\beta|$. In addition, the path $\mathcal{C}$ can only touch the cone whenever the eigenvalues of $M_{\text{cell}}$, namely $e^{ik_B a}$, are degenerate. This is necessarily the case at the edges of the Brillouin zone ($k_B = \pm\frac{\pi}{a}$), and at its center $k_B = 0$. In between, $\mathcal{C}$ cannot touch the cone, since two distinct eigenvalues $e^{\pm ik_B a}$ must be found, by virtue of time-reversal symmetry. Finally, the path $\mathcal{C}$ is not a loop, but a simple line, since $M_{\text{cell}}$ is a simple function of $\omega$, and therefore is the same for two opposite values of $k_B$ on a band: it starts on the cone at $k_B = -\frac{\pi}{a}$ and lands on it again at $k_B = 0$, before following the reverse path between $k_B = 0$ and $k_B = \frac{\pi}{a}$. Figure 6a represents an example of $\mathcal{C}$ contour for the third band of the crystal (supposedly topologically "trivial" case,

with $e_p = 2.8$ cm), and Fig. 6c represents the same contour for $e_p = -2.8$ cm, corresponding to the dual system, which is supposedly topological (the topological properties will be proven in the next section). Figure 6b represents the case $e_p = 0$ cm that closes the bandgaps. As expected, in all cases the contour starts and end on the cone.

To study the conditions under which two consecutive frequency bands can touch, it is convenient to recast the Bloch eigenproblem into the equivalent form:

$$e^{-ik_B a} M_{\text{cell}}(\omega)|\psi\rangle = |\psi\rangle \tag{22}$$

and think of it as follows: for each $k_B$ in the first Brillouin zone, finding the bands means finding the values of $\omega$ for which the matrix $e^{-ik_B a} M_{\text{cell}}$ has at least one eigenvalue equal to one, with the corresponding eigenvector being the Bloch eigenvector on that particular band. This can happen for infinitely many values of $\omega$. If both eigenvalues of $e^{-ik_B a} M_{\text{cell}}$ at a given frequency are equal to one, the band structure is doubly degenerate, which is therefore the maximum frequency degeneracy allowed by the system. Since the general form of the eigenvalues of

$e^{-ik_B a} M_{\text{cell}}$ on a band are $v_\pm = e^{-ik_B a}\left(\alpha_R \pm i\sqrt{\alpha_I^2 - |\beta|^2}\right) = e^{-ik_B a}e^{\pm ik_B a}$, the

second eigenvalue $e^{-2ik_B a}$ can only become equal to unity at the Brillouin zone edges ($k_B = \pm\frac{\pi}{a}$), or at $k_B = 0$. As a consequence, bandgaps can only close at the center or edge of the Brillouin zone, i.e. when the contour $\mathcal{C}$ touches the cone.

Assuming the first case, i.e. a degeneracy at $k_B = \pm\frac{\pi}{a}$, one has $e^{-ik_B a} = -1$. We obtain, at the particular frequency of the degeneracy,

$$e^{-ik_B a} M_{\text{cell}} = \begin{pmatrix} 1 \mp i|\beta| & -\beta* \\ -\beta & 1 \pm i|\beta| \end{pmatrix} \tag{23}$$

and this matrix can only be equal to identity if $|\beta| = 0$. The second case of degeneracy at $k_B = 0$ leads to the same conclusion ($|\beta| = 0$). This means that when

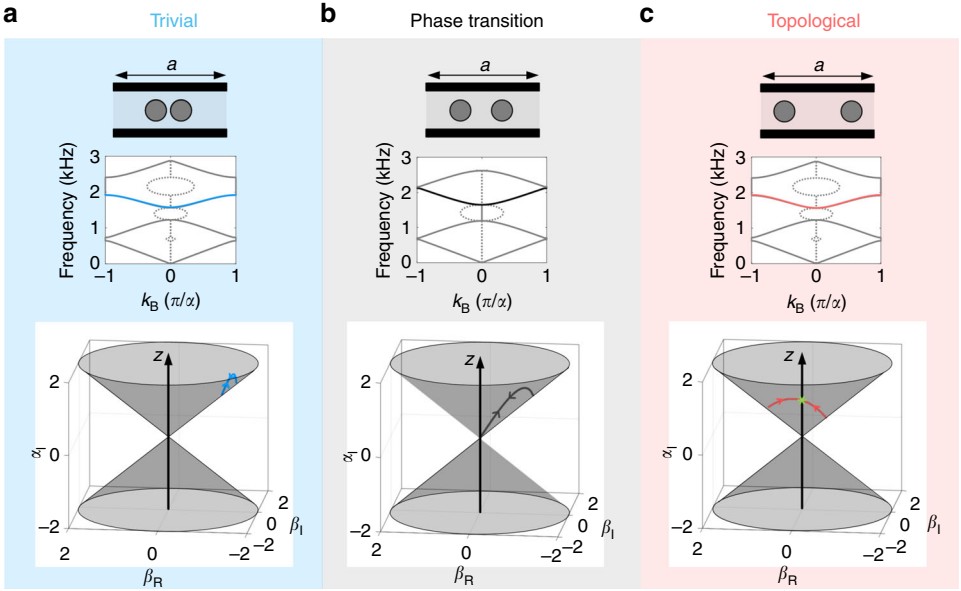

**Fig. 6 Topology of the bands.** We define the topology of the bands as the number of times the contours $\mathcal{C}$ crosses the axis of the cone defined in Eq. 20. **a** For the trivial lattice, the contour $\mathcal{C}$ does not cross the axis of the cone, corresponding to a zero topological invariant. **b** When the system goes through phase transition, the contour $\mathcal{C}$ touches the tip of the cone. The topological invariant cannot be defined in this case. **c** Same as panels (**a**) and (**b**) but for the topological lattice. The contour $\mathcal{C}$ crosses the axis of the cone one time in this case, which corresponds to a nontrivial topology

two bands touch, the contour $\mathcal{C}$ is reaching the tip of the cone, as confirmed by Fig. 6b.

**Topology of the bands.** As seen in previous sections, each band defines a mapping between the Brillouin circle and a subspace of SU(1,1) matrices. We now define a topological invariant for each band, i.e. an integer quantity that is invariant upon continuous transformations of the band structure. This means that this number can only change when the band undergoes a discontinuous transformation, i.e. touches another one, or equivalently when the contour $\mathcal{C}$ touches the tip of the cone.

Like in the standard tight-binding SSH model, we need an extra symmetry, akin to chiral symmetry, to be able to define topological invariants on each band. Here we need to require that the scattering matrices $S_1$ and $S_2$ are equal, taking $\theta_1 = \theta_2 = \theta$, $\alpha_{1,2} = \alpha_{1,2} = \alpha$ and $\varphi_1 = \varphi_2 = \varphi$. With this extra condition, the quantity $\beta = M_{21}(\omega(k_B))$ in Eq. 14, that parametrizes the matrix $M_{cell}$ on a band, becomes

$$\beta(k_B) = -2e^{i(\varphi - \alpha)} \cos\left(\alpha + \frac{\omega(k_B)d}{c}\right) \cot\theta \csc\theta, \tag{24}$$

where the quantities $\alpha$, $\theta$ and $\varphi$ that parametrize the $S$ matrix of a single obstacle generally depend on $\omega(k_B)$. We then assume the case of nonresonant scatterers, meaning that $\cos\theta$ does not vanish on the band, and the variation of $\alpha$ and $\theta$ on the band are negligible. Because $M_{cell}$ always has two complex-conjugate unimodular eigenvalues, $\omega(k_B)$ is necessarily monotonous between $-\pi/a$ and 0. Let us focus our attention to the quantity $\cos\left(\alpha + \frac{\omega(k_B)d}{c}\right)$, which can potentially make the complex number $\beta(k_B)$ vanish at some particular point of the Brillouin zone. When $k_B$ goes from $-\pi/a$ to 0, the angle $\gamma = \alpha + \frac{\omega(k_B)d}{c}$ moves monotonically between two real values, say $\gamma_{min}$ and $\gamma_{max}$, defining a continuous monotonous mapping between $\left[-\frac{\pi}{a}, 0\right]$ to $[\gamma_{min}, \gamma_{max}]$. Now, two situations can arise:

(1)   The segment $[\gamma_{min}, \gamma_{max}]$ does not contain $\pi/2$ (modulo $\pi$), in which case $\cos\left(\alpha + \frac{\omega(k_B)d}{c}\right)$ never vanishes as $k_B$ go from $-\pi/a$ to 0. This means that $\beta$ never vanishes on the band.

(2)   The segment $[\gamma_{min}, \gamma_{max}]$ contains $\pi/2$ (modulo $\pi$), in which case $\beta$ vanishes at least once on the band.

Since $\beta = 0$ means that the contour $\mathcal{C}$ crosses the cone axis, we can therefore define a topological invariant $\eta$ in the following way: We can count the number of times $\eta$ that $\mathcal{C}$ crosses the cone axis as $k_B$ goes from $-\pi/a$ to 0. This integer number changes each time $\gamma_{max}$ or $\gamma_{min}$ equals $\pi/2$ (modulo $\pi$), i.e. when $\beta$ is zero either at the edge or center of the Brillouin zone, i.e. when a band gap closes. Figure 6 shows how the contour $\mathcal{C}$ evolves for the third band of our system, when one goes from the trivial regime (panel a, $\mathcal{C}$ does not cross the cone axis, $\eta = 0$) to the topological one (panel c, $\mathcal{C}$ crosses the cone axis, $\eta = 1$). At the topological phase transition, the contour $\mathcal{C}$ touches the tip of the cone, which closes the band gap, and the number $\eta$ is not defined.

**Symmetry protection.** The definition of the topological invariant $\eta$ as the number of times the contour $\mathcal{C}$ crosses the cone axis between $-\pi/a$ to 0 is based on two underlying symmetries, and both must be fulfilled:

1.   (1) Time-reversal symmetry, which guarantees that $M_{cell}$ belongs to SU(1,1)[55].

1.   (2) Equality of $S_1$ and $S_2$ (the far-field individual scattering matrices of both obstacles must be identical), or equivalently:

$$M_{cell}^2 = 1. \tag{25}$$

Obviously, horizontal position disorder does not change the individual scattering parameters of the object. In addition, vertical position disorder does not change it either, as demonstrated in Supplementary Fig. 12 (the only difference in the scattering spectrum are very sharp Fano interferences occurring from coupling to a acoustic bound state in the continuum, but they are far from the frequency range of interest). As a consequence, position disorder does not break $M_{cell}^2 = 1$. However, changing the diameter of one rod definitely changes its scattering matrix. What happens in the case of rods with different radii is that the real and imaginary part of the quantity

$$\beta(k_B) = -e^{\frac{i\omega(k_B)d}{c}} e^{i\varphi_2} e^{i(a_1 - a_2)} \csc\theta_1 \cot\theta_2 - e^{\frac{i\omega(k_B)d}{c}} e^{i\varphi_1} e^{-i(a_1 + a_2)} \cot\theta_1 \csc\theta_2 \tag{26}$$

are never simultaneously zero, which implies that the contour $\mathcal{C}$ can avoid crossing the cone axis by simply going around it. This is analogous to a SSH chain without chiral symmetry, where some properly chosen chirality-breaking defects at an interface can change the winding number without closing the band gap. These results explain the outcome of the full-wave simulations presented in Fig. 3 of the main text.

**Numerical methods.** Full-wave simulations are all performed using Comsol Multiphysics (Acoustic and RF modules). Dispersion curves are obtained by considering a single unit cell of the lattice arrays, applying Floquet boundary condition to the lateral sides of the unit cell, and performing eigenfrequency simulations for all of the Floquet−Bloch wavenumbers.

In order to obtain the frequency spectra of the ODE solvers, we excite the system with an incident plane-wave with unit amplitude and measure the amount of pressure at the transmission side of the waveguide.

In order to cross-validate our experimental measurements, we performed numerical finite-element simulations including a viscothermal loss of 1.15 dB/m to achieve a transfer function $X(\omega)$, for example, between the injected and transmitted sound waves. We then obtained the transfer function of the loudspeaker $Y(\omega)$ by exciting the empty waveguide and measuring the associated sound pressure level at the transmission side. The transfer function $Z(\omega)$, between the voltage applied to the loudspeaker and the transmitted pressure, was then readily obtained as $Z(\omega) = X(\omega)/Y(\omega)$.

In our FDTD simulations, we excite the waveguide from one end with the desired modulated input signal, and record the temporal evolution of the pressure field (with a time step subject to Courant–Friedrichs–Lewy (CFL) condition for ensuring stability) received at a point on the other side of the waveguide.

**Experimental methods**. As mentioned in the main text, an acrylic square tube is used to implement the acoustic waveguide. Nylon 6 continuous cast cylinders were then manually inserted into the waveguide to form the SSH-type array. Supplementary Fig. 13a represents the experimental setup used to achieve the transfer function of the system. The setup contains a loudspeaker, a Data Physics Quattro signal analyzer connected to a computer (not shown in the figure) controlling it, one ICP microphone measuring the transmitted sound pressure level, and a home-made anechoic termination (not shown in the figure). To obtain the transfer function of the sample, we drive the loudspeaker with a burst noise voltage (which is set as the reference signal in the setup), and measure the pressure level with respect to the reference channel using the ICP microphone. Supplementary Fig. 13b shows the experimental setup used to create an input signal (voltage) with an arbitrary time profile $\tilde{g}(t)$, and to measure the temporal evolution of the output signal $\tilde{f}(t)$. The setup consists of a Speedgoat Performance Real-Time Target Machine with IO131 interface controlled by xPC target environment of MATLAB/Simulink, a loudspeaker, a power amplifier, a home-made acoustic termination (not shown in the figure), and an ICP microphone measuring the transmitted pressure.

## Data availability

The datasets generated during and/or analyzed during the current study are available from the corresponding author on reasonable request.

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

## Acknowledgements

This work was supported by the Swiss National Science Foundation (SNSF) under Grant No. 172487. The authors would like to thank Dr Nadège Kaina for useful discussions about topological properties of multiple scattering systems.

## Author contributions

F.Z-N. carried out the research work under the supervision of R.F. Both authors participated in discussing the results and writing the manuscript.

## Additional information

**Competing interests:** The authors declare no competing interests.

