## [Peer Review File · Nature Communications]

Reviewers' comments:

Reviewer #1 (Remarks to the Author):

In this work, the author demonstrated theoretically and experimentally a two-port topological insulator resonator based on the SSH model for analog signal processing. The demonstrated acoustic analog signal processor (APS) is functionally robust against disorder and geometrical defects. Consequently, the reliability and flexibility of these kinds of ASPs can be improved for future large-scale device applications.

It is an interesting work to bring the concept/advantage of topology phononic/photonic crystal to the APS field. The analysis in the manuscript have been carried out concisely, also the results are scientifically sound. I recommend it for publication if the following issues are excluded.

1) In their manuscript, the author mentioned twice the term "on-the-fly", which is not clear to me. The author should explain it clearly or cite references.

2) "while repeating the same operation always gives rise to the same result when using DSPs, analog signal processing is often accompanied with considerable observational error caused by the extreme sensitivity of ASPs to changes in environmental and structural parameters [21]." —The logic of this sentence is not rigorous. Do these changes (in environmental and structural parameters) have no effect on the DSP but only ASP? This should be discussed in more details.

3) References are not adequate, for example, topological insulator resonator in "Elastic pseudospin transport for integratable topological phononic circuits, Nature Communications 9 (1), 3072". Also, the author should prove reference for the following part "According to coupled-mode theory, the transmission coefficient..." for the readers to follow their work in depth.

4) Can the author comment more on their disorders in Fig.2 and Fig.3? Is 18% the upper limit of the robustness in their system? What will a greater degree of disorder bring? Also, have the authors considered other types of defects?

5) It seems to me that the author used fewer rods in the experiment (Fig.3) than in the numerical simulation (Fig.2). Can the author also comment on this?

Reviewer #2 (Remarks to the Author):

In this manuscript, the authors explore the possible application of topological concepts to wave-based analog signal processing, with the aim of mitigating the sensitivity of the response with respect to disorder and geometrical perturbations.

Within this framework, they study a proof-of-concept platform which, via resonant tunneling through a defected photonic bandgap material, can analogically "solve" linear differential equations of arbitrary order in the time domain as wave signal propagates through it. In particular, they show that such "trivial" configuration is significantly affected by geometrical imperfections, and that this sensitivity is strongly reduced when the resonant transmission phenomenon is mediated by a topological edge mode. Such evidence is ascertained via numerical simulations and acoustic experiments.

The manuscript is generally well written, and the results look interesting and technically sound. In particular, the joint application of two emerging concepts like topological effects and wave-based analog computing seems original, timely and with potentially broad impact.

Nevertheless, I feel that some important issues need to be addressed before the manuscript can be recommended for publication.

1. I feel that the proposed approach is not properly contextualized within the state of the art, which makes it difficult to assess the actual depth and broadness of its impact. Almost all references cited in the introduction connection with analog signal processing (Refs. 4-14) deal with space-domain (rather than time-domain) processing of the wavefront. The only one that seems directly comparable (solving linear differential equations in the time domain) is that in Ref. 13,

which exploits a single silicon microring resonator. Therefore, two questions naturally arise:

a) Could the proposed topological concepts be beneficial for space-domain processing as well? This is an important issue since, as it also emerges from the bibliography referenced, it would have a much broader impact.

b) In connection with the time-domain processing considered, one might question to what extent the sensitivity to perturbations is inherent of the specific photonic-crystal-based platform chosen by the authors. Since it all boils around attaining a suitable filtering response, could a different design be inherently more immune to disorder, even without resorting to topological concepts? I am thinking, for instance, of the approach in Ref. 13 (single silicon microring resonator) or designs based on metatronics concepts which have been demonstrated capable of synthesizing transfer functions with fairly general filtering properties. These approaches do not rely on order-sensitive concepts like bandgaps, and may be inherently more robust to imperfections.

2. It is not clear to me why the numerical and experimental demonstrations (Figs. 2 and 3 respectively, as well as Fig. 4, and Figs. S4-S6) pertain to different configurations, and so cannot be directly compared. I believe that such direct comparison would be useful also for cross-validation. Moreover, it looks like the trivial configuration chosen for the numerical simulations is much more sensitive to perturbations than the experimental counterpart. I would like the Authors to elaborate on these issues.

Minor issues

- It is not clear to me why Ref. 14 is cited in connection with "equation solving".
- After Eq. (1), the time-harmonic convention utilized should be explicitly defined.
- On p. 12, "less number of" perhaps should read "a smaller number of".
- There seem to be some glitches in the cross-referencing. For instance, Figs. S4 and S5 should refer to Fig. 2 (not Fig. 3) and Fig. S6 should refer to Fig. 3 (not Fig. 4).

Reviewer #3 (Remarks to the Author):

This paper proposes the use of topologically protected modes for analog signal processing. The main proposition is that these modes can be robust to disorder, and therefore signal processing using these modes could potentially be more robust. The authors provide a few demonstrations using acoustic topological structures. I find this idea intriguing but I do not believe that this paper is ready for publication, as I explain below.

The idea that resonators (or a network of coupled resonances) can be used to produce complicated and high-order transfer functions is something that undergraduates have studied for decades. This is the foundation of any basic course on signal processing or controls. The authors call this "ODE solving", but at its core it is just basic filter design. Renaming this well-established science is cute but not fundamentally new. The statement from the abstract "In particular, we achieve *advanced* [emphasis mine] signal processing tasks, such as resolution of linear differential equations" is not accurate.

The major change, however, that is proposed here is that instead of using an explicit resonator (e.g. microwave or optical or acoustic cavity, microwave stripline resonators, L-C tanks, MEMS resonators), the authors just change the resonator type to a topological defect resonator. They then argue that these resonators can be more robust than their conventional counterparts. As a comparative case, they use a phononic crystal resonator and show that disorder disturbs its resonances. In my opinion this choice of comparison is arbitrary and should not be the focus of the study. Moreover, the authors do not even discuss how more advanced filter operations could be set up using ladders / networks / arrays of coupled topological defect modes, so the discussion and demonstrations remain very basic.

As a secondary point, the authors do not mention anywhere in the paper how they set the resonance frequencies and what influences them. It is known that SSH edge-modes are protected by chiral symmetry and are sensitive to on-site potential variations – though it is not at all clear how this works exactly in their system since they don't give any information. Since they use a symmetry-protected topological mode (and it is not time-reversal symmetry or something that is difficult to break), any disorder that breaks the symmetry should also break their device. Unfortunately, there is no mention at all that the modes are symmetry protected (if fact they never even use the word symmetry) and on how the disorder is set in their demo.

There is a major missed opportunity here. The authors could have just said that filter design is an established and advanced science. They could have then argued that replacing the poles/zeros of a filter through topological defects has merit. They could have then methodically and rigorously quantified the robustness of complicated transfer functions produced using topologically protected resonator modes, instead of just showing a couple of rudimentary examples. It is not clear to me how exactly they could build this case rigorously and quantitatively, but I recommend the authors consider this when reworking this paper.

Reviewer #4 (Remarks to the Author):

In this manuscript, the authors propose taking advantage of the robustness inherent to topological insulators as a way to overcome the sensitivity to disorder in analog signal processors. Although digital signal processors are ubiquitous nowadays, the authors make a compelling case for the use of analog signal processors, particularly if robustness can be achieved. Regardless of its potential for practical use, I find the discovery of this robustness of fundamental interest. I think this work may open the venue for further explorations that unify the field of topological insulators with the field of signal processing and linear system theory.

However, I have one main concern:

The results clearly show that some form of robustness exists when disorder is introduced into the topological signal processor. This is in striking contrast with the distortion observed for the case of a non-topological processor. However, the mechanism of protection is not explained. From topological band theory, it is known that a mid-gap state will localize at the interface between SSH chains of opposite winding number and that this state is protected against disorder as long as chiral symmetry is preserved. In this work, this mid-gap energy is used as the resonance for the topological signal processor. On the other hand, the non-topological processor uses "a resonance induced by defect-tunneling through a Bragg band gap". In both cases, in-gap resonances are used. The robustness, however, does not pertain the existence of the resonance itself since in both cases a resonance is maintained in the presence of disorder. Rather, the robustness is in the response of the signal as it passes through the resonance. This crucial issue is not addressed by the authors, but it is at the heart of the topological protection they show. The manuscript is therefore incomplete from the theory side. If a thorough mechanism is not intended to be provided, at least the authors should discuss/comment on the mechanism of protection.

In relation to the topological band theory described in the Supplementary Information file, the existence of zero energy states is protected by chiral symmetry, and the topological invariant is the winding number (see for example Teo and Kane, Phys. Rev. B, 82, 115120 (2010)). The Zak phase does not protect the zero energy state. This section should be corrected, as well as any other mention of the Zak phase as being the protecting invariant.

Finally, since the kind of topological protection the authors claim exists only in the presence of certain symmetries, a discussion on the symmetries protecting the robustness is lacking. To test that the robustness is due to the symmetry protected topological phase, the system should be tested against disorder that (i) breaks the protecting symmetries (not shown) and (ii) do not break

the protecting symmetries (shown).

I would like to see these issues addressed before making a final recommendation regarding publication.

We would like to thank all of the Referees for their time and careful review of our manuscript. We highly appreciate the Referees' keen interest in our work, as evident from their detailed comments about our manuscript as "an interesting" (quoted from Referee A) and "intriguing" (quoted from Referee C) study with "potentially broad impact" (quoted from Referee B), which "opens the venue for further explorations that unify the field of topological insulators with the field of signal processing and linear system theory" (quoted from Referee D). In the following we provide a detailed response to the comments and recommendations of all of the Referees, and point out to all the changes made, which are highlighted in yellow in the revised manuscript and supplementary file.

Referee A

We appreciate the Referee's sharp interest in our work and thank him/her for the careful review of our manuscript, and his/her positive opinion. In what follows, we address the issues raised by the Referee.

Comment A1: In their manuscript, the author mentioned twice the term "on-the-fly", which is not clear to me. The author should explain it clearly or cite references.

Response: With the term "On-the-fly" we meant that our proposed equation solver carries out the computation in real time, without any delay for storage or conversion.

Revision: We have rephrased the sentences containing this expression in the revised manuscript.

Comment A2: "while repeating the same operation always gives rise to the same result when using DSPs, analog signal processing is often accompanied with considerable observational error caused by the extreme sensitivity of ASPs to changes in environmental and structural parameters [21]." The logic of this sentence is not rigorous. Do these changes (in environmental and structural parameters) have no effect on the DSP but only ASP? This should be discussed in more details.

Response: Environmental and structural variations may also affect DSPs in some cases. However, there exist quite advanced error-finding algorithms and protocols that substantially reduce the observational errors of DSPs, making them much more reliable and flexible than their analog counterparts.

Revision: We have commented on this point in the revised manuscript so as to make the quoted sentence perfectly clear and logic (see page 3, paragraph 1).

Comment A3: References are not adequate, for example, topological insulator resonator in "Elastic pseudospin transport for integratable topological phononic circuits, Nature Communications 9 (1), 3072". Also, the author should prove reference for the following part "According to coupled-mode theory, the transmission coefficient..." for the readers to follow their work in depth.

Revision: We have cited the mentioned work in the revised manuscript (see Ref. 45). Besides, we have provided a proper reference for the mentioned part (Ref. 51).

Comment A4: Can the author comment more on their disorders in Fig.2 and Fig.3? Is 18% the upper limit of the robustness in their system? What will a greater degree of disorder bring? Also, have the authors considered other types of defects?

Response: Like any other topological system, the robustness of the proposed equation solver is ultimately limited by Anderson localization. More specifically, the proposed equation solver works well as long as the applied disorder is not strong enough so that it closes the surrounding topological band-gaps.

To address the second part of your comment, we have thoroughly discussed and analyzed the effect of disorders of different types on the topological equation solver. This is summarized in the new Figure 3, and explained by the theory added in Methods.

Revision: We have commented on these points in the manuscript (see Fig. 3 of the revised manuscript and the discussion added in pages 6, 7, and Methods).

Comment A5: It seems to me that the author used fewer rods in the experiment (Fig.3) than in the numerical simulation (Fig.2). Can the author also comment on this?

Response: You are correct, we used fewer rods in the experiment in order to work with a smaller sample. However, we understand the point of the Referee that a direct comparison between numerical simulations and measurements must be included in the manuscript.

Revision: We have performed a new numerical simulation, which corresponds to the exact situation of the experiment and takes into account the viscothermal losses and the dynamics of the excitation. The obtained numerical results are represented in Figure 4 of the revised manuscript (grey colored curves), showing perfect agreement with the measurements.

Referee B

We appreciate the Referee's interest in our work and thank him/her for the careful review of our manuscript and his/her positive opinion. In what follows, we address the issues raised by the Referee.

Comment B1: I feel that the proposed approach is not properly contextualized within the state of the art, which makes it difficult to assess the actual depth and broadness of its impact. Almost all references cited in the introduction connection with analog signal processing (Refs. 4-14) deal with space-domain (rather than time-domain) processing of the wavefront. The only one that seems directly comparable (solving linear differential equations in the time domain) is that in Ref. 13, which exploits a single silicon microring resonator. Therefore, two questions naturally arise:

- a) **Could the proposed topological concepts be beneficial for space-domain processing as well? This is an important issue since, as it also emerges from the bibliography referenced, it would have a much broader impact.**

Revision: We have now cited more works related to analog signal processing in time domain (Refs. 25-30), showing that this is an important direction as well. About generalizing the concept to the spatial domain, although a complete study of such an extension is beyond the scope of the present work, we briefly discuss such a possibility in the supplementary material, section V. We also comment on this in the main text.

- b) **In connection with the time-domain processing considered, one might question to what extent the sensitivity to perturbations is inherent of the specific photonic-crystal-based platform chosen by the authors. Since it all boils around attaining a suitable filtering response, could a different design be inherently more immune to disorder, even without resorting to topological concepts? I am thinking, for instance, of the approach in Ref. 13 (single silicon microring resonator) or designs based on metatronics concepts which have been demonstrated capable of synthesizing transfer functions with fairly general filtering properties. These approaches do not rely on order-sensitive concepts like bandgaps, and may be inherently more robust to imperfections.**

Response: While single resonators can be inherently immune to disorder, they clearly do not offer a very large range of filtering responses. If one wants to achieve more advanced processing tasks (similar to the higher order filtering responses discussed in the manuscript), the usage of several of these building blocks is inevitable. The key question in such a platform is then similar to the one in our multiple scattering crystal: how can we mitigate the effect of disorder on the coupling between these building blocks or on their frequency tuning conditions, in order to preserve the overall response function? Topology is an interesting solution because, in contrast to trivial resonance systems which are sensitive to vanishingly small disorder strengths, topological systems can be immune to certain classes of defects as long as the disorder strength is not large enough to close the bandgap. We believe this is a quite unique feature that deserves to be pointed out and exploited.

Regarding the second potential platform mentioned in the comment, namely metatronics, we do not believe that there is a necessity to oppose it to what we propose. Metatronics is basically optical circuit theory, that is to say a tool to design, by assembling lumped optical circuit element, a desired optical response. This tool can be used to create transfer functions, in conjunction or not with topological concepts. Our approach is of course, not based on optimization, as is often the case in metatronic designs, however the combination of topological and metatronic concepts may allow one to design sturdy response functions with an increased control over the transfer function, while benefitting from a topological immunity to various classes of defects.

Revision: We now discuss the possibility of combining metatronics and topology for enhancing the robustness and flexibility of analog signal processing (see the discussion section).

Comment B2: It is not clear to me why the numerical and experimental demonstrations (Figs. 2 and 3 respectively, as well as Fig. 4, and Figs. S4-S6) pertain to different configurations, and so cannot be directly compared. I believe that such direct comparison would be useful also for cross-validation. Moreover, it looks like the trivial configuration chosen for the numerical simulations is much more sensitive to perturbations than the experimental counterpart. I would like the Authors to elaborate on these issues.

Response: There are three major causes for the difference observed between our numerical and experimental demonstrations:

1- We used fewer rods in our experiment. This was in order to avoid working with a very large experimental setup.
2- In the simulation results shown in Fig. 2, we have neglected the acoustic losses that are inevitably present in the experiment.

3- In our simulation (Fig. 2), the input signal is the sound wave injected into the waveguide and the output is the sound pressure level received at the other side of the waveguide. However, in our measurement, the input signal is the *voltage* applied to the loudspeaker and the sound pressure level measured by the microphone at the transmission side is the output. Hence, one has to consider the transfer function of the loudspeaker as well so that the experimental and simulation results become comparable with each other.

Revision: In the revised manuscript, we have performed extra simulations with the same number of rods, including viscothermal losses and the dynamic of the excitation (see Methods). The obtained numerical results are now in perfect agreement with the measurements (see Fig. 4, 5 and S3).

Comment B3: It is not clear to me why Ref. 14 is cited in connection with “equation solving”.

Revision: We have replaced this mistakenly cited reference. We meant to cite another one, now Ref. 17 of the revised manuscript.

Comment B4: After Eq. (1), the time-harmonic convention utilized should be explicitly defined.

Revision: We fixed this issue in the revised manuscript.

Comment B5: On p. 12, “less number of” perhaps should read “a smaller number of”.

Revision: We have fixed it.

Comment B6: There seem to be some glitches in the cross-referencing. For instance, Figs. S4 and S5 should refer to Fig. 2 (not Fig. 3) and Fig. S6 should refer to Fig. 3 (not Fig. 4).

Revision: We proofread the whole text one more time and fixed the mentioned glitches in the cross-referencing.

Referee C

We are thankful to Referee C for his/her careful review of our work and appreciate the various comments and recommendations. In what follows, we address the issues raised by the Referee.

Comment C1: The idea that resonators (or a network of coupled resonances) can be used to produce complicated and high-order transfer functions is something that undergraduates have studied for decades. This is the foundation of any basic course on signal processing or controls. The authors call this “ODE solving”, but at its core it is just basic filter design. Renaming this well-established science is cute but not fundamentally new. The statement from the abstract “In particular, we achieve *advanced* [emphasis mine] signal processing tasks, such as resolution of linear differential equations” is not accurate.

Response: We appreciate the comment of the Referee. It is our purpose to target a simple signal processing task that even undergraduates can grasp in order to evidence in a simple way the relevance of topological concepts in signal processing. We agree that the proposed topological equation solvers can also be viewed as robust frequency filters, and can be equally understood as such. However, a choice must be made in the presentation, and since our signal processing operations and measurements are performed in time-domain, we find it more relevant to talk about differential equation solving. In fine, these two visions are equivalent, and we are glad to point this out in our manuscript.

Revision: We removed the term “advanced” in the abstract, and pointed out the equivalence between frequency filters and time-domain analog signal processors (see page 9 of the main text, and supplementary materials, section VI).

Comment C2: The major change, however, that is proposed here is that instead of using an explicit resonator (e.g. microwave or optical or acoustic cavity, microwave stripline resonators, L-C tanks, MEMS resonators), the authors just change the resonator type to a topological defect resonator. They then argue that these resonators can be more robust than their conventional counterparts. As a comparative case, they use a phononic crystal resonator and show that disorder disturbs its resonances. In my opinion this choice of comparison is arbitrary and should not be the focus of the study.

Response: We agree with the Referee that comparing the trivial phononic resonance to the topological one should not be the entire focus of the paper. In fact, many other aspects like the origin of the topological protection, which is not obvious in such multiple-scattering systems, and symmetry protection, are also very important. We have added a substantial part in the Methods to clarify the topology of the system, and demonstrated that it is protected by a special symmetry of the transfer matrix. This also led us to analyze in a more general fashion the types of defects that the system is immune to, and draw a clearer and more rigorous picture of what can be reasonably achieved with topological signal processing.

Revision: We have explained the topological theory of the system in the Methods section. We have further discussed the underlying symmetry that protects the edge mode of the proposed system. The main text is also updated with a new Figure 3.

Comment C3: Moreover, the authors do not even discuss how more advanced filter operations could be set up using ladders / networks / arrays of coupled topological defect modes, so the discussion and demonstrations remain very basic.

Response: We had demonstrated, both in theory and experiment, how adding the responses of two topological resonances allows one to achieve a second-order filtering response, corresponding to the transfer function of a second order ODE (see Fig. 5 of the revised manuscript). Furthermore, we had discussed in the method section how the transfer functions of arbitrary orders can be constructed by adding or subtracting the responses of different topological resonances to or from each other. We agree, however, that this is not the only solution for achieving advanced filter operations. An alternative strategy, mentioned by the Referee, is to allow the topological resonators to couple to each other.

Revision: We have discussed and demonstrated this possibility in the revised manuscript (see page 9, and supplementary materials section VI).

Comment C4: As a secondary point, the authors do not mention anywhere in the paper how they set the resonance frequencies and what influences them. It is known that SSH edge-modes are protected by chiral symmetry and are sensitive to on-site potential variations - though it is not at all clear how this works exactly in their system since they don't give any information. Since they use a symmetry-protected topological mode (and it is not time-reversal symmetry or something that is difficult to break), any disorder that breaks the symmetry should also break their device. Unfortunately, there is no mention at all that the modes are symmetry protected (if fact they never even use the word symmetry) and on how the disorder is set in their demo.

Response: In the revised manuscript, we have clarified the topological properties of the system and its underlying symmetry protection. In regular tight-binding SSH chains, made of evanescently coupled identical resonators, the mid-gap edge mode occurring at the topological boundary is protected by chiral symmetry. Hence, a transfer function based on tunneling through this edge mode is robust to disorder in the hoppings, as long as they are weak enough not to close the band gap. Our multiple scattering system, albeit not based on evanescent coupling, behaves similarly. The transmission peak of the ordered sample survives disorder shifts that do not close the band gap, but not disorder in the obstacle radii. To explain these results, we have provided the theory for the topology of the system based on the transfer matrix of the unit cell M_{cell} . This theory proves that the edge mode of the system is protected by the symmetry $M_{cell}^2 = 1$, which holds for the position disorder but not for the radii disorder, as we have demonstrated in Methods.

Revision: We have explained the topological theory of the system in the Methods section. We have further discussed the underlying symmetry that protects the edge mode of the proposed system. The main text is also updated with a new Figure 3.

Comment C5: There is a major missed opportunity here. The authors could have just said that filter design is an established and advanced science. They could have then argued that replacing the poles/zeros of a filter through topological defects has merit. They could have then methodically and rigorously quantified the robustness of complicated transfer functions produced using topologically protected resonator modes, instead of just showing a couple of rudimentary examples. It is not clear to me how exactly they could build this case rigorously and quantitatively, but I recommend the authors consider this when reworking this paper.

Response and revisions: There are certainly different ways to present the results and we have made the choice of what we believe to be a wide-audience approach, which talks about simple differential equations and has the advantage of fitting well with the time-domain experiments performed. We demonstrate the case of first-order equations, then second order ones, and provide a potential route to extend the concept to arbitrary equations (i.e. arbitrary transfer functions). Following the Referee's advice, we now also discuss the possibility of achieving more complex filtering responses making use of the topological resonances supported by cascaded SSH arrays (see supplementary materials, section VI). Furthermore, we methodically and rigorously quantified the robustness of the corresponding topological resonating modes from statistical averages, and commented on the results in relation with the symmetry arguments demonstrated in Methods.

We think that these changes have led to a consistent and complete story and hope that the Referee will appreciate the way that the revised manuscript is now presented.

Referee D

We appreciate the Referee's sharp interest in our work and thank him/her for the careful review of our manuscript and his/her very positive opinion. In what follows, we have addressed the issues raised by the Referee.

Comment D1: The results clearly show that some form of robustness exists when disorder is introduced into the topological signal processor. This is in striking contrast with the distortion observed for the case of a non-topological processor. However, the mechanism of protection is not explained. From topological band theory, it is known that a mid-gap state will localize at the interface between SSH chains of opposite winding number and that this state is protected against disorder as long as chiral symmetry is preserved. In this work, this mid-gap energy is used as the resonance for the topological signal processor. On the other hand, the non-topological processor uses "a resonance induced by defect-tunneling through a Bragg band gap". In both cases, in-gap resonances are used. The robustness, however, does not pertain the existence of the resonance itself since in both cases a resonance is maintained in the presence of disorder. Rather, the robustness is in the response of the signal as it passes through the resonance. This crucial issue is not addressed by the authors, but it is at the heart of the topological protection they show. The manuscript is therefore incomplete from the theory side. If a thorough mechanism is not intended to be provided, at least the authors should discuss/comment on the mechanism of protection.

Response: In the revised manuscript, we have clarified the topological properties of the system and its underlying symmetry protection. In regular tight-binding SSH chains, made of evanescently coupled identical resonators, the mid-gap edge mode occurring at a topological boundary is protected by chiral symmetry. Hence, a transfer function based on tunneling through this edge mode is robust to disorder in the hoppings, as long as they are weak enough not to close the band gap. Our multiple scattering system, albeit not based on evanescent coupling, behaves similarly. The transmission peak of the ordered sample survives disorder shifts that do not close the band gap, but not disorder in the obstacle radii. To explain these results, we have provided the theory for the topology of the system based on the transfer matrix of the unit cell M_{cell} . This theory proves that the edge mode of the system is protected by the symmetry $M_{cell}^2 = 1$, which holds for position disorder but not for the radii disorder, as we have demonstrated in Methods.

Revision: We have added the full topological theory of the system in the Methods section. We have further discussed the underlying symmetry that protects the edge mode of the proposed system. The main text is also updated with a new Figure 3.

Comment D2: In relation to the topological band theory described in the Supplementary Information file, the existence of zero energy states is protected by chiral symmetry, and the topological invariant is the winding number (see for example Teo and Kane, Phys. Rev. B, 82, 115120 (2010)). The zak phase does not protect the zero energy state. This section should be corrected, as well as any other mention of the Zak phase as being the protecting invariant.

Response: We agree. The new added theory defines the topological invariant in a rigorous way, which makes this point no longer relevant.

Comment B3: Finally, since the kind of topological protection the authors claim exists only in the presence of certain symmetries, a discussion on the symmetries protecting the robustness is lacking. To test that the robustness is due to the symmetry protected topological phase, the system should be tested against disorder that (i) breaks the protecting symmetries (not shown) and (ii) do not break the protecting symmetries (shown).

Response and revisions: This is done in the newly added Figure 3, and discussed in the accompanying text (page 6 and 7 of the revised manuscript). The Methods fully discusses the symmetry protection of the edge modes.

* * *

REVIEWERS' COMMENTS:

Reviewer #1 (Remarks to the Author):

The authors have addressed all comments and suggestions. In my opinion, the manuscript is now in suitable form for publication.

Reviewer #2 (Remarks to the Author):

The authors have satisfactorily addressed the issues that I raised. I have no further comments.

Reviewer #3 (Remarks to the Author):

I don't really have any further comments. I still find the work to be incremental and non-transformative.

Reviewer #4 (Remarks to the Author):

the authors have addressed my concerns in a satisfactory manner. In particular, they have detailed the theoretical framework that was lacking in the original manuscript and have provided experimental evidence of the symmetry protection mechanism by adding different types of disorder to the system. As a result, the paper is in much better shape now.